# Unsupervised Gait Event Identification with a Single Wearable Accelerometer and/or Gyroscope: A Comparison of Methods across Running Speeds, Surfaces, and Foot Strike Patterns

**DOI:** 10.3390/s23115022

**Published:** 2023-05-24

**Authors:** Dovin Kiernan, Kristine Dunn Siino, David A. Hawkins

**Affiliations:** 1Biomedical Engineering Graduate Group, University of California, Davis, Davis, CA 95616, USA; dahawkins@ucdavis.edu; 2Department of Neurobiology, Physiology and Behavior, University of California, Davis, Davis, CA 95616, USA

**Keywords:** inertial measurement units, in-field sport and athlete monitoring, initial contact, heel strike, toe off, stance, swing, step

## Abstract

We evaluated 18 methods capable of identifying initial contact (IC) and terminal contact (TC) gait events during human running using data from a single wearable sensor on the shank or sacrum. We adapted or created code to automatically execute each method, then applied it to identify gait events from 74 runners across different foot strike angles, surfaces, and speeds. To quantify error, estimated gait events were compared to ground truth events from a time-synchronized force plate. Based on our findings, to identify gait events with a wearable on the shank, we recommend the Purcell or Fadillioglu method for IC (biases +17.4 and −24.3 ms; LOAs −96.8 to +131.6 and −137.0 to +88.4 ms) and the Purcell method for TC (bias +3.5 ms; LOAs −143.9 to +150.9 ms). To identify gait events with a wearable on the sacrum, we recommend the Auvinet or Reenalda method for IC (biases −30.4 and +29.0 ms; LOAs −149.2 to +88.5 and −83.3 to +141.3 ms) and the Auvinet method for TC (bias −2.8 ms; LOAs −152.7 to +147.2 ms). Finally, to identify the foot in contact with the ground when using a wearable on the sacrum, we recommend the Lee method (81.9% accuracy).

## 1. Introduction

In this paper, we explore the accuracy and precision of different methods to identify gait events during human running using a single wearable device (accelerometer, gyroscope, or inertial measurement unit). Identifying when the foot contacts and leaves the ground—the initial contact (IC) and terminal contact (TC) gait events, respectively—allows biomechanists to break the cyclic movements of running into discrete phases. Identifying IC and TC events allows the analysis of basic but highly useful temporal variables like stride (right IC to right IC or left IC to left IC), step (right IC to left IC or left IC to right IC), stance/contact (right IC to right TC or left IC to left TC), swing (right TC to right IC or left TC to left IC), and flight/float (right TC to left IC or left TC to right IC) times and frequencies (1/time) (Figure 1). These basic temporal variables are critical in the analysis and prediction of running speed and performance [1,2,3,4]. Segmenting running into these discrete phases also forms the starting point for many more advanced biomechanical analyses that quantify and compare kinematic or kinetic features’ means and variances. Thus, accurately identifying IC and TC events is a critical first step in the analysis and evaluation of running.

IC and TC events can be identified with relative ease using force plates or instrumented treadmills in labs. Further, several methods have been developed to identify these events from lab-based motion capture [5,6,7,8]. However, lab settings impose constraints on data collection: force plates, instrumented treadmills, and motion capture systems require participants to visit a lab, are expensive, and require a trained technician. These ‘captive’ systems may also cause participants to alter their gait (e.g., treadmill or short running track) and limit the volume of data collection to a few gait cycles [9,10,11,12]. Thus, ‘captive’ systems only provide ‘snapshots’ of data, decreasing ecological validity and limiting investigations of phenomena that occur across many gait cycles such as adaptation, fatigue, movement variability, and overuse injury [13,14,15,16,17,18,19,20]. Advances in technology such as instrumented insoles have allowed biomechanists to overcome some of these constraints but suffer from their own issues with durability and comfort [21]. Consequently, researchers remain largely reliant on ‘captive’ technology.

The emergence of small, low-cost accelerometers, gyroscopes, and inertial measurement units (IMUs) (collectively called ‘wearables’ for the remainder of this paper) offers a new way forward [22]. These portable sensors can continuously record data during prolonged runs in natural environments. Wearables have high rates of adoption among runners [23] and have demonstrated accuracy in measuring or estimating several biomechanical parameters, including gait events [22,24,25,26]. Based on these promising findings, our objective was to identify and evaluate wearable-based methods of gait event identification that met two criteria: (1) the method needed to minimize system complexity by relying on a single wearable located on the shank or sacrum/low-back (in contrast to sets of multiple wearables); and (2) the method needed to be automated, using an unsupervised processing code to prevent bottlenecks caused by manually processing thousands or millions of gait events. 

In line with this objective, we identified 18 candidate methods (Table 1). As inputs, candidate methods used accelerations from the shank (8 of 18) or sacrum (5 of 18), angular velocities from the shank (3 of 18) or sacrum (1 of 18), or both accelerations and angular velocities from the shank (1 of 18). Several (5 of 18) heavily filtered or otherwise manipulated their inputs using a variety of techniques (multi-resolution analysis, complementary signals, over-smoothing, wavelet-mediated differentiation, Principal Component Analysis). The majority (17 of 18) used rules- or threshold-based heuristic approaches to estimate events (e.g., find signal max, then find zero-crossing preceding max), while one used a machine learning approach (Echo State Network). Further descriptions of individual methods are provided in the Appendix A.

Importantly, each of these methods represents a promising approach for in-field gait event identification, but unfortunately, comprehensive comparisons of these methods have not been conducted, leaving users with little guidance on selecting the best method for their application. Further, most methods have at least one of the following limitations: Most methods were developed and validated (1) using a small sample (*n* ≤ 10: [27,28,29,30,31,32,33,34,35]; 11 ≤ *n* ≤ 20: [36,37,38,39,40,41,42]; *n* > 20: [43,44]), (2) using a narrow range of, or unreported, running speeds (single speed: [32,36,37,39,43], speeds not reported: [28,30,31,35,38,40]), (3) using a single running surface (all except [44]), (4) without accounting for foot-strike kinematics (foot strike not reported: [27,28,30,31,33,34,35,36,38,40,41,43]; only one strike pattern studied: [29,32,37,39]), and (5) without adequate synchronization between wearable data and a ground truth reference (no synchronization reported: [27,31,36,39,43]; imprecise synchronization: [33,41,42,44]). These limitations decrease the confidence with which we can apply these wearable-based methods broadly across conditions and participants. 

To address these limitations and help users make an informed decision about which of the many available methods to use for their application, we attempted to replicate each of the 18 methods using code published by the authors (2 of 18), by acquiring code directly from the authors (2 of 18), or by following descriptions in the original papers to produce new code (14 of 18). We also created modified versions of three methods, resulting in code for 21 methods (code for each method is available at https://github.com/DovinKiernan/REID_IMU_Running_Event_ID, published on 3 May 2023). We then evaluated each method to determine how well it could identify IC and TC events from a large sample that was collected across a range of running speeds, surfaces, and foot strike angles. To do so, we quantified: (1) how often each method failed to identify gait events; (2) the method’s accuracy (bias); (3) precision (error variance); (4) the effects of speed, surface, and foot strike angle on accuracy; and (5) the time required to execute each method.

**Table 1 sensors-23-05022-t001:** Summary of methods to identify gait events from running using data from a single wearable on the shank or sacrum/lower back. M—male; F—female; NR—not reported; FF—forefoot; RF—rearfoot; SS—self-selected; α—acceleration; ω—angular velocity; SCS—segment coordinate system; WCS—wearable coordinate system; GCS—global coordinate system (coordinate conventions defined below). IC—initial contact; TC—terminal contact; RL—right/left.

Sensor Location	Method	Sample	Foot-Strike	Speed	Surface	Placement	Signals	Sampling Frequency	Events	Ground Truth	Sync
Shank	Mizrahi [36]	*n* = 14 (14 M) Healthy	NR	3.5 ± 0.2 m/s	Treadmill	Tibial tuberosity	$a_{WCS,Y}$	1667 Hz	IC	None	N/A
	Mercer [27]	*n* = 10 (10 M) Healthy	NR	3.1–3.8 m/s	Treadmill	Anteromedial distal tibia	$a_{WCS,Y}$	1000 Hz	IC, TC	None	N/A
	Purcell [28]	*n* = 6 Healthy	NR	SS jog, run, and sprint	Overground	Anteromedial distal tibia	$a_{WCS,X,Y,\&Z}$	250 Hz	IC, TC	Forceplate 1000 Hz	TTL pulse
	Greene/McGrath [29,45]	*n* = 5 (4 M; 1 F) Healthy	RF	0.6–3.3 m/s	Treadmill	Anterior aspect of mid shank	$\omega_{WCS,Z}$	102.4 Hz	IC, TC	MoCap 200 Hz	TTL pulse
	Aminian/ O’Donovan [35,46]	*n* = 1 (1 M) Healthy	NR	SS jog	Overground	Shank	$\omega_{SCS,Z}$	102.4 Hz	IC, TC	MoCap 200 Hz	TTL pulse
	Sinclair [37]	*n* = 16 (11 M; 5 F)	RF	4.0 ± 0.2 m/s	Overground	Anteromedial distal tibia	$a_{WCS,Y}$	1000 Hz	IC, TC	Forceplate 1000 Hz	Synchronous recording
	Whelan [30]	*n* = 7 (3 M; 4 F) National and international sprinters	NR	≤50% max effort	Overground	Anteromedial mid-tibia	$a_{WCS,X}$	148.2 Hz	IC	Forceplate 1000 Hz	TTL pulse
	Norris [31]	*n* = 6 (1 M; 5 F) Recreational	NR	SS half-marathon training	Overground	Anteromedial distal tibia	$a_{WCS,Z}$	204.8 Hz	IC	None	N/A
	Schmidt [38]	*n* = 12 (10 M; 2 F) Track and field athletes	NR	SS sprint	Overground	Lateral distal tibia	$a_{WCS,Y}$ $\omega_{WCS,Z}$	1000 Hz	IC, TC	Photocell	NR
	Aubol [39]	*n* = 19 (9 M; 10 F) ≥16.1 km/wk Injury free	RF	3.0 ± 0.2 m/s	Overground	Anteromedial distal tibia	$a_{WCS,X,Y,\&Z}$	1000 Hz	IC	Forceplate 1000 Hz	Synchronous recording
	Fadillioglu [40]	*n* = 13 (13 M) Injury free	NR	SS walking and running	Overground	Leg	$\omega_{WCS,Z}$	1500 Hz	IC, TC	Forceplate 1000 Hz	TTL pulse
	Bach [43]	*n* = 21 (13 M; 8 F) Healthy	NR	2.2 ± 0.1 m/s	Treadmill	Anteromedial proximal tibia	$a_{WCS,X,Y,\&Z}$	142.9 Hz	IC, TC	Forceplate 1000 Hz	NR
Sacrum/Lower back	Auvinet [32]	*n* = 7 (7 M) “top-level”	RF	5.2 ± 0.1 m/s	Overground	Lumbar spine	$a_{WCS,X,Y,\&Z}$	100 Hz	IC, TC, RL	MoCap 200 Hz	Photoflash
	Lee [33]	*n* = 10 (6 M; 4 F) National standard runners	NR	2.8–5.3 m/s	Treadmill	Sacrum (S1)	$a_{WCS,X\&Z}$	100 Hz	IC, TC, RL	MoCap 100 Hz	Vertical movement
	Wixted [34]	*n* = 2 Nationally ranked	NR	5.9–6.2 m/s	Overground	Lumbar spine (L3–L4)	$a_{WCS,X\&Y}$	500 Hz	IC, TC	Insoles 500 Hz	Synchronous collection
	Bergamini [41]	*n* = 11 (7 M; 4 F) Amateur and national track and field team	NR	5.7–10.8 m/s	Overground	Lumbar spine (L1)	$\omega_{WCS,X,Y,\&Z}$	200 Hz	IC, TC	Forceplate/Mocap 200 Hz/300 Hz	Hammer tap/ none
	Benson [44]	*n* = 54 (29 M; 25 F) Recreational	FF and RF	2.7–3.6 m/s	Treadmill and Overground	Lower back	$a_{WCS,X,Y,\&Z}$	201 Hz	IC, TC, RL	Forceplate 1000 Hz	Vertical jump
	Reenalda [42]	*n* = 20 (15 M; 5 F) ≥15 km/week; no injuries	FF and RF	3.1–4.2 m/s	Treadmill	Sacrum	$a_{GCS,Y}$	240 Hz interpolated to 1000 Hz	IC	Forceplate 1000 Hz	x-correlated MoCap

## 2. Materials and Methods

### 2.1. IMU Calibration

Tri-axial IMUs (ProMove MINI, Inertia Technology, Enschede, The Netherlands; ±16 g primary, ±100 g secondary, ±34.91 rad/s, 1000 Hz) were secured to a centrifuge (ClearPath MCVC, Teknic, Victor, NY, USA) with custom 3-D printed jigs (SOLIDWORKS 2019, Dassault Systèmes, Vélizy-Villacoublay, France) and calibrated in 6 orientations at 16 known accelerations (from 0 to 41.42 g, where 1 g = 9.8 m/s^2^ [47,48]) and angular velocities (from 0 to 78.54 rad/s). Adapting methods from Coolbaugh et al. [49], known data (K) from the centrifuge and measured data (M) from the IMU were used to calculate 3 × 7 calibration matrices (C; 3 signed magnitude terms, 3 absolute magnitude terms, and one bias term per axis) and quantify sensor accuracy after subtracting out biases observed during a quiet period (B) (Equation (1)). After calibration, IMU primary accelerometer errors were ≤0.01 ± 0.04 g, secondary accelerometer errors were ≤0.05 ± 0.07 g, and gyroscope errors were ≤0.01 ± 0.01 rad/s.
C ∗ (M + B) = K(1)

### 2.2. Participants

Seventy-seven participants ≥18 years old who reported running ≥16 km per week for ≥6 months were recruited from the University of California, Davis, local running clubs, and the community at large. Three participants were excluded from analysis due to movement of an IMU (*n* = 2) or inability to complete the protocol as instructed (*n* = 1), leaving a final sample of 74 (32 males; 42 females; age 28 ± 12 years; Figure 2). All participants provided written informed consent, and procedures were approved by the University of California, Davis Institutional Review Board.

### 2.3. Protocol

Using adhesive-bonded hook-and-loop fasteners, IMUs were attached to neoprene belts with anti-slip silicone inners, then wrapped with elastic straps as tightly as possible, within the limit of participant comfort. IMUs were placed anterior and superior to the lateral malleoli, on the superior aspect of the iliac crests in line with the greater trochanter, and on the superior aspect of the sacrum in line with the spine (Figure 3). Only data from tibial and sacral IMUs are analyzed here.

Participants wore their own shoes and ran a 25 m runway with an embedded force plate (Kistler 9281, Kistler Group, Winterthur, Switzerland; 1000 Hz). Running speed was recorded using two custom-built laser speed gates, placed 2.5 m on each side of force plate center. Participants warmed up and practiced striking the force plate three times per side at their *slowest* (“the slowest pace you would use on a run”), *typical* (“the pace you use for the majority of your running”), and *fastest* (“the fastest pace you would use on a run”) self-selected speeds. During this warmup, markers on the lateral calcaneus and base of the fifth metatarsal were recorded using a conventional camera (Exilim EX-FH25, Casio; 120 Hz). Foot strike angle was calculated by subtracting a neutral standing foot angle (Kinovea 0.9.5) with positive values indicating a more dorsiflexed foot at IC and values > 0.14 rad corresponding to rear-foot strike, −0.03 to 0.14 rad to mid-foot strike, and <−0.03 rad to forefoot strike [51]. After warm-up, five stances per side were collected at each speed for two surface conditions: (1) with a track surface covering the runway and force plate (*track*) and (2) with no covering on the hardwood floor of a basketball court (*floor*). Participants always progressed from their slowest to fastest speeds, but the order of foot and surface was pseudo-randomized.

IMU data were synchronized within 100 ns of each other with a wireless network hub (Advanced Inertia Gateway, Inertia Technology, Enschede, The Netherlands; 1000 Hz). This hub periodically sent voltage pulse trains that were synchronously recorded by IMUs and a custom MATLAB script that simultaneously recorded the speed and force data (R2018b, MathWorks, Natick, MA, USA). Pulse trains were cross-correlated to synchronize signals.

### 2.4. Data Processing

Quiet periods were identified (angular velocity < 0.5 rad/s and jerk < 0.01 m/s^3^ for at least 100 ms) and used to remove biases from IMU data. Saturated frames from the primary accelerometer ($\left|a\right|$ > 15.5 g) were replaced with corresponding frames from the secondary accelerometer. Data were filtered with a 4th-order 50-Hz low-pass Butterworth filter. Angular velocity was drift-corrected using a Madgwick filter [52,53,54]. Starting at each quiet period, accelerations were used to estimate IMU position in the inertial reference frame, then angular velocities were used to estimate frame-by-frame changes in IMU orientation and remove the gravity component from accelerations [55]. Data were then expressed in a segment coordinate system based on the Principal Component that explained the most variance in angular velocity during running (the medial-lateral axis) and the gravity vector during quiet standing [56,57]. Force plate data were filtered with a 4th-order 50-Hz low-pass Butterworth filter. A vertical force threshold of 10 N was used to define the start and end of stance (IC and TC, respectively). IMU data were segmented using the time the participant crossed the first and second speed gates, yielding 5 m of running data for each trial. For more detailed IMU processing, see Appendix A.

During data processing, we observed small timing discrepancies caused by the initialization of discrete MATLAB data acquisitions and small variances in the sampling rates of the IMU and MATLAB systems. Although extremely small, these discrepancies could accumulate over the course of the ~60 min data collection, leading to timing differences between the first and last synch events of a data collection (on the order of 10 s of ms). Given this study’s focus on event timing, a conservative approach was used, and only trials that contained a synch event were analyzed (642 of 4440 trials). All other trials were discarded to ensure millisecond-level accuracy was maintained.

### 2.5. Analysis

To quantify how well each method agreed with ground truth gait events (IC and TC) measured by the force plate, error was quantified for each of the 642 trials with each method. The error was calculated by subtracting method-estimated gait-event timings from force plate gait-event timings. To account for the non-independence of the data (642 trials from 74 participants) and ensure proper estimation of variance, IC and TC errors were entered into two separate linear mixed-effects models in R (v4.2.2; R Foundation for Statistical Computing, Indianapolis, IN, USA) as described in Carstensen et al.’s approach to linked replicates (Equation (2)) [58]:(2)$$y_{method,participant,trial}=a_{method}+b_{participant}+c_{participant,trial}+d_{method,participant}+e_{method,participant,trial}$$

This approach allowed us to quantify variance components and estimate: (1) method biases (accuracy); (2) limits of agreement (LOA) within which 95% of future errors for a given method are expected to fall; and (3) within-method standard deviations that quantify method repeatability (precision) (see also [59,60,61]). Model assumptions of independence, normality, and homoscedasticity were validated by plotting within-participant variances against within-participant means, histograms of residuals, residuals for each level of random effect, and residuals as a function of fitted value.

To evaluate if any potential explanatory variables affected method error, a second set of linear mixed effects models was developed for each method. These models added surface, speed, and foot strike angle as fixed effects. A threshold of *p* ≤ 0.05 was used to evaluate whether a fixed effect explained a significant amount of a given method’s error.

Finally, to quantify the processing time required to execute each method, the same computer (Intel Core i9-13900HX; Kingston Fury 64 GB DDR5 5600 MT/s; Gammix S70 Blade 7400 MB/s SSD) was used to process two 30 min, 1.8 million frame (1000 Hz), steady state running trials obtained from a separate study (but using the same IMUs, placement, and pre-processing) [62]. Each trial was processed 100 times per method, and processing times were recorded.

## 3. Results

### 3.1. Failure to Identify Gait Events or Step Side

Some methods were unable to recognize any gait events within the 5 m of running that was segmented for analysis in each trial: Aminian/O’Donovan_modified missed 18.8% of trials, Greene/McGrath_modified missed 0.9%, Whelan missed 1.6%, Norris missed 7.3%, Schmidt missed 63.9%, Aubol missed 6.4%, Fadillioglu missed 1.1%, and Bach missed 27.5% (Figure 4). All other methods identified gait events in every trial.

Methods using wearables placed on the shank only provided information ipsilateral to placement, and thus the side contacting the ground during stance was always known. In contrast, methods using wearables placed on the sacrum/low-back provided information on bilateral gait events. To account for this, three of six sacrum/low-back placed methods could identify the side (i.e., right or left) contacting the ground. Of these, the Lee method was the most accurate, correctly identifying sides 81.9% of the time. The Benson and Auvinet methods correctly identified sides 54.6 and 75.0% of the time (Figure 5).

### 3.2. Initial Contact

Using Carstensen’s method for linked replicates [58], biases, within-method SDs, and 95% LOAs (1.96 SD of errors) for the IC estimation of each method were modeled (Figure 6). Results revealed high biases and/or LOAs for several methods; thus, a second figure displaying only the best-performing methods (LOAs within ±200 ms) is also provided (Figure 7). The best-performing methods for shank-mounted wearables were Mizrahi (+18.6 ms bias, −112.2 to +149.3 ms LOA), Mercer (−31.7 ms bias, −166.3 to +102.9 ms LOA), Purcell (+17.4 ms bias, −96.8 to +131.6 ms LOA), and Fadillioglu (−24.3 ms bias, −137.0 to +88.4 ms LOA). The best-performing methods for sacrum/low-back mounted wearables were Auvinet (−30.4 ms bias, −149.2 to +88.5 ms LOA), Lee (+29.1 ms bias, −92.9 to +151.0 ms LOA), Wixted (+29.6 ms bias, −92.5 to +151.7 ms LOA), Benson (+19.2 ms bias, −129.6 to +168.1 ms LOA), and Reenalda (+29.0 ms bias, −83.3 to +141.3 ms LOA).

We performed a second set of linear mixed effects models on each method to examine the role of potential explanatory variables. These models revealed that none of the methods’ IC estimations were significantly affected by surface (all *p*s > 0.05). In contrast, speed explained a significant amount (*p* ≤ 0.05) of the error in 3 of 15 shank-placed methods and 5 of 6 sacrum-/low-back-placed methods, and foot strike angle explained a significant amount of the error in 1 of 15 shank- and 0 of 6 sacrum-/low-back-placed methods. Table 2 displays the coefficients for each potential explanatory variable and whether the explanatory variable significantly explained the method error. To better illustrate these effects, model-predicted mean absolute errors were also plotted as a function of speed and foot strike angle (Figure 8 for the best-performing methods; Appendix A for others). “Hotter” yellow colors in these plots correspond to higher predicted mean absolute error (MAE), while “cooler” blue colors correspond to lower predicted MAE. Predicted MAEs are plotted as a function of the speeds (*x*-axis) and foot strike angles (*y*-axis) observed in this study. For example, the vertical bands of cooler colors in the Wixted, Benson, and Lee plots suggest that these methods work well at a certain range of speeds but have higher predicted MAE outside of those ranges, while the hotter colors in the corners of the Mercer and Reenalda graphs suggest a combination of slower speeds and high (Mercer) or low (Reenalda) foot strike angles increases predicted MAE. In contrast, the relatively consistent colors in the Mizrahi, Purcell, Fadillioglu, and Auvinet graphs suggest the predicted MAE is stable across the range of foot strike angles and speeds modeled here.

### 3.3. Terminal Contact

Using Carstensen’s method for linked replicates [58], biases, within-method SDs, and 95% LOAs for each method’s TC estimation were calculated (Figure 9). Results revealed high biases and/or LOAs for several methods; thus, a second figure displaying only the best-performing methods (LOAs within ±200 ms) is also provided (Figure 10). The best-performing methods for shank-mounted wearables were Purcell (+3.5 ms bias, −143.9 to +150.9 ms LOA) and Fadillioglu (+17.7 ms bias, −148.9 to +184.4 ms LOA). The best-performing methods for sacrum/low-back mounted wearables were Auvinet (−2.8 ms bias, −152.7 to +147.2 ms LOA), Lee (+24.3 ms bias, −139.4 to +187.9 ms LOA), Wixted (−43.8 ms bias, −192.7 to +105.1 ms LOA), and Benson (+26.6 ms bias, −130.4 to +183.5 ms LOA).

We performed a second set of linear mixed effects models on each method to examine the role of potential explanatory variables. These models revealed that none of the methods’ TC estimations were significantly affected by surface or foot strike angle (all *p*s > 0.05), while speed explained a significant amount (*p* ≤ 0.05) of error in 5 of 11 shank-placed methods and 3 of 5 sacrum-/low-back-placed methods. Table 2 displays the coefficients for each potential explanatory variable and whether the explanatory variable significantly explained the method error. To better illustrate these effects, model-predicted MAEs were plotted as a function of speed and foot strike angle (Figure 11 for the best-performing methods; Appendix A for others). These plots suggest that predicted MAEs for the Purcell, Auvinet, and Wixted methods are relatively stable across speeds and foot strike angles. In contrast, the Lee method has an increase in predicted MAE at lower speeds, the Benson method has an increase in predicted MAE at higher speeds and lower foot strike angles, and the Fadillioglu method has an increase in predicted MAE at lower speeds and higher foot strike angles.

### 3.4. Processing Time

Finally, to quantify the time required to execute each method, two 30 min, 1.8 million-frame, steady-state running trials were processed 100 times with each method. Processing times were recorded and plotted in Figure 12. Most methods were able to process data in <5 s, all methods in <20 s.

## 4. Discussion

We identified 18 separate methods to estimate initial contact (IC) and terminal contact (TC) gait events from running using a single wearable sensor on the shank or sacrum/low back. We modified three of these original methods in an attempt to improve performance, resulting in a total of 21 methods. For each method, we either adapted (4 of 18) or created (14 of 18) code to automatically process data (available at https://github.com/DovinKiernan/REID_IMU_Running_Event_ID published on 3 May 2023). We then used each automated method to estimate gait events from 74 runners across two different surfaces (wood floor and running track), three self-selected speeds (slowest, typical, and fastest), and foot strike angles (ranging from forefoot to rearfoot strike patterns). To quantify error, these estimated gait events were compared to ground truth force plate events.

Overall, we found errors to be higher than reported in the original studies (except [42,44]). This is likely a function of several factors: First, in contrast to some studies, we time synchronized our ground truth force plates with our IMUs with ms accuracy. Second, to our knowledge, the current study represents the largest sample used to validate and compare running gait event identification methods. Most studies used samples of 11 ± 13 participants (mean ± SD) to both develop and validate their method. Third, the current study used a range of conditions, including different surfaces, speeds, and foot strike angles. In contrast, the methods studied were developed and validated under a narrow set of conditions (except [44]). Thus, errors reported for most studies are likely a function of both developing and validating on the same small sample of participants and narrow set of conditions. When adopting a gait event identification method, users should consider the conditions and participants with which the method was developed and validated and not assume that the method will work under other conditions or for other participants. 

For example, O’Donovan et al. [35] and McGrath et al. [29] adapted methods developed for walking [45,46] without specifying any changes to address differences between walking and running. These methods had high errors, often mislabeling gait events from pre-/proceeding steps as the event of interest due to their large time windows. On the other side of the spectrum, Schmidt et al. [38] developed their method specifically for sprinting. This likely explains the high error observed across the range of running speeds we used here. As a final example, Bach et al. [43] trained an Echo State Network to estimate gait events using data from a narrow set of conditions. To replicate their model, we used the full running data set published with their original paper but still observed high errors when estimating gait events from our data. This likely stems from the model being applied to conditions on which it was not trained. It is possible that further training in these novel conditions could improve results.

These observations should also be applied to the current study, given that we only quantified the error for over-ground running on two level surfaces. It should not be assumed that our results would hold for incline/decline running, treadmill running, or running on other surfaces (e.g., sand, grass/turf, asphalt, concrete). That said, none of the evaluated methods had significantly different errors across the two surfaces we used, suggesting these methods could be used on similar surfaces (e.g., concrete, asphalt). Finally, all methods studied here have been developed and validated on a relatively homogenous group of runners. Thus, if studying participants drawn from different populations, our results may not be representative. For example, individuals with lower extremity amputation could have higher frequency components in their data that could disrupt pattern recognition [63], or data from individuals with more subcutaneous body fat could contain more noise from soft tissue artifacts [64,65].

With those caveats in mind, for IC estimation with a shank-placed wearable, we recommend either the Purcell [28] or Fadillioglu [40] methods. Both methods were able to identify gait events in almost all trials (100% and 98.9%) and stood out with low biases (+17.4 and −24.3 ms) and LOAs (−96.8 to +131.6 and −137.0 to +88.4 ms). Further, neither method was significantly affected by running surface, speed, or foot strike angle, suggesting that both methods could be applied broadly across conditions.

For IC estimation with a sacrum or low-back placed wearable, we recommend the Auvinet [32] or Reenalda [42] methods. Both methods were able to identify gait events in 100% of trials and stood out with low biases (−30.4 and +29.0 ms) and LOAs (−149.2 to +88.5 and −83.3 to +141.3 ms). The Auvinet method was not significantly affected by running surface, speed, or foot strike angle. In contrast, the Reenalda method was significantly affected by running speed, with errors increasing at low running speeds as well as a trend toward errors increasing at lower foot strike angles. Thus, the Reenalda method may be preferable when speeds are known to exceed ~4 m/s while the Auvinet method may be preferable for speeds below ~4 m/s or when conditions are unknown. The Lee, Wixted, and Benson methods [33,34,44] also present viable options when speed is known to fall within certain ranges (see Figure 7).

For TC estimation with a shank-placed wearable, we recommend the Purcell method [28]. This method identified gait events in 100% of trials with low bias (+3.5 ms) and LOAs (−143.9 to +150.9 ms). Error was not significantly affected by surface or foot strike angle but was significantly affected by speed. The slope for speed was, however, quite low, indicating that error only changed a small amount across speeds (4.00 ms per 1 m/s).

For TC estimation with a sacrum-/low-back-placed wearable, we recommend the Auvinet method [32]. This method identified gait events in 100% of trials with low bias (−2.8 ms) and LOAs (−152.7 to +147.2 ms). The error for TC estimation with the Auvinet method was not significantly affected by surface or foot strike angle but was significantly affected by speed. At speeds below ~5 m/s, the Lee method [33] could also be used. Although the Lee method’s bias (+24.3 ms) and LOAs (−139.4 to 187.9 ms) were higher overall, this appears to be driven largely by the significant effect of speed, with lower speeds causing larger errors.

To identify the side with a sacrum-/low-back-placed wearable, we recommend the Lee method [33], which correctly identified the side in 81.9% of gait events. Given these high success rates and the predictable left-right-left-right pattern of running, the probability of misidentifying gait events should decrease exponentially as a function of the number of steps, quickly reaching negligible values. To illustrate, based on the number of steps observed in the 5 m trials used here, we observed a probability of misidentification of 0.18. With another 5 m of steps, we would again have a 0.18 probability of misidentification. Therefore, we can roughly estimate that the probability of misidentifying all steps in 10 m should be 0.18^2^ or 0.03. Over 1 km, this would fall to roughly 0.18^(1000 m/5 m)^ or 1 × 10^−149^. Thus, over longer data collections, the chances of misidentifying the side a gait event is occurring on become extremely small.

As seen in the recommendations above, users should balance the reported accuracy of a method against the potential negative effects that running speed and foot strike angle could have on it. One approach to address these concerns and improve accuracy may be to include speed and foot strike angle as model inputs. For example, Patoz et al. [66] and Alcantara et al. [67] provided their contact time estimation models with running speed data and reported impressively low error. Unfortunately, these methods estimate the *mean* contact times across a data set rather than the timing of individual gait events. Despite this limitation, these methods illustrate the potential benefits of including speed as a model input. Speed is relatively easy to calculate from GPS, and several methods have already been proposed to estimate speed and foot strike angle from accelerations or angular velocities [68,69,70]. Thus, the inclusion of speed, foot strike angle, and other potential explanatory variables may be a relatively easy way to improve the performance of future methods and should be a target of additional research.

With the continued development of biomechanics data collection and processing methods that can be used both in the lab and in the field, biomechanics will become increasingly accessible. Although increasing accessibility was a key goal of this study (through minimizing system complexity and data processing supervision), this increase in accessibility also brings challenges. For example, when deploying wearables in the field, they may be placed by individuals with little training. This could lead to misalignment and degrade accuracy. To mitigate this issue, we used a segment coordinate system (SCS). In contrast, most methods studied here (16 of 18) originally used wearable coordinate systems (WCSs) defined by the axes of the sensors within the wearable. Using a WCS is less reliable due to (1) differences in wearable manufacturing (sensor axes may not perfectly align with their housing or each other), (2) participant geometry (e.g., tibial or sacral morphology may differ across participants), and/or (3) wearable placement (e.g., wearable may be placed upside down or at an ‘improper’ angle). Thus, we recommend (1) calibrating the wearable to ensure the output is accurately expressed in a consistent coordinate system [49] and (2) creating a SCS based off (a) the gravity vector during quiet standing and (b) the first Principal Component calculated from calibration motions such as rotation about the medial-lateral axis (e.g., leg swings or inverted pendulum about the ankles) [56,57], and/or (c) the gravity vector while lying prone/supine. Using this approach, even if a wearable is ‘misaligned,’ the SCS will be unaffected. These or similar methods will help mitigate the potential pitfalls of deploying wearables, particularly when deployed in the field. Although we believe using a SCS is best practice, for the sake of comparison, we also executed our analyses on data expressed in both a WCS and a pseudo-global “tilt-corrected” coordinate system (see Appendix A). In line with a priori expectations, these analyses show that—even though the same trained experimenter placed every wearable in this study—MAEs for the recommended methods were, on average, 59.4 ms higher with a WCS than a SCS. 

## 5. Conclusions

For wearables on the shank, we recommend the Purcell [28] or Fadillioglu [40] method to identify IC and the Purcell method to identify TC. For wearables on the sacrum/low-back, we recommend the Auvinet [32] or Reenalda [42] method to identify IC, the Auvinet method to identify TC, and the Lee [33] method to identify the side in contact with the ground. These methods are accurate and precise across the speeds and foot strike angles observed in this study (roughly 2.5–7.5 m/s and −0.25–0.75 rad) and across level track and hardwood surfaces. We also recommend that input data be expressed in a SCS rather than a WCS. Future work should validate these methods across surfaces with different durometers and inclines/declines. Future work could also improve the accuracy and precision of IC and TC estimation by combining techniques from the methods examined here or by developing new methods. Novel methods may benefit from including additional information (e.g., running speed or foot strike angle). As seen by the 18 methods examined here, the field is saturated with options for running gait event identification. This saturation is problematic given that direct comparison between methods has been rare [31,71], making it difficult for users to identify the best method for their application. Thus, future work should use the methods recommended here as a benchmark for comparison when developing new methods.

## Figures and Tables

**Figure 1 sensors-23-05022-f001:**
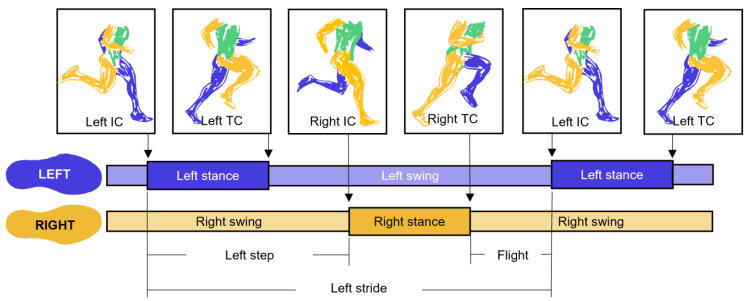
Gait phases defined by initial contact (IC) and terminal contact (TC) gait events. Left arm and leg represented in blue; right arm and leg represented in orange; trunk represented in green.

**Figure 2 sensors-23-05022-f002:**
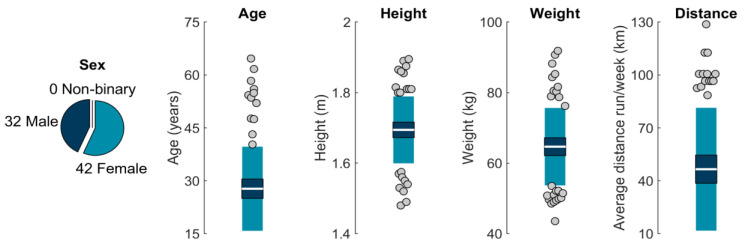
From left to right: sample sex, age, height, weight, and self-reported average distance run per week. The white horizontal line represents the mean; dark blue represents ±95% confidence interval (±1.96 SEM) around the mean; and light blue represents ±1 SD around the mean. Gray dots represent participants outside ±1 SD.

**Figure 3 sensors-23-05022-f003:**
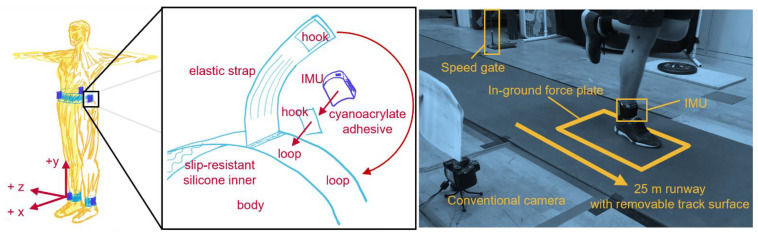
**Left**: IMU placement and coordinate conventions. For consistency, different conventions used across methods have been standardized to ISB conventions [50]: Segment coordinate systems (SCS) were defined as anterior (+x), proximal (+y), and medial-lateral (with right defined as +z); wearable coordinate systems (WCS) were defined square to the IMU housing, which was roughly aligned with the direction of progression (+x), longitudinal axis (+y), and right (+z). **Middle**: Belt design and IMU fixation. **Right**: Experimental setup.

**Figure 4 sensors-23-05022-f004:**
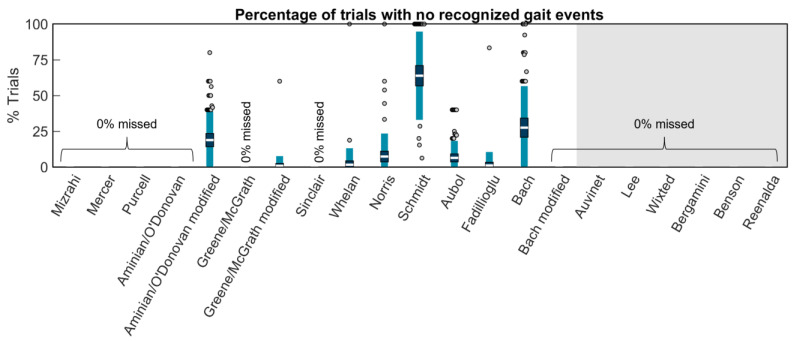
The white horizontal line represents the mean percentage of trials per participant without any gait events recognized; dark blue represents ±95% confidence interval (±1.96 SEM) around the mean; and light blue represents ±1 SD around the mean. Gray dots represent participants outside ±1 SD. No bars indicate that gait events were identified in every trial for every participant. Methods on the white background are for wearables on the shank. The methods on the gray background are for wearables on the sacrum/low back.

**Figure 5 sensors-23-05022-f005:**
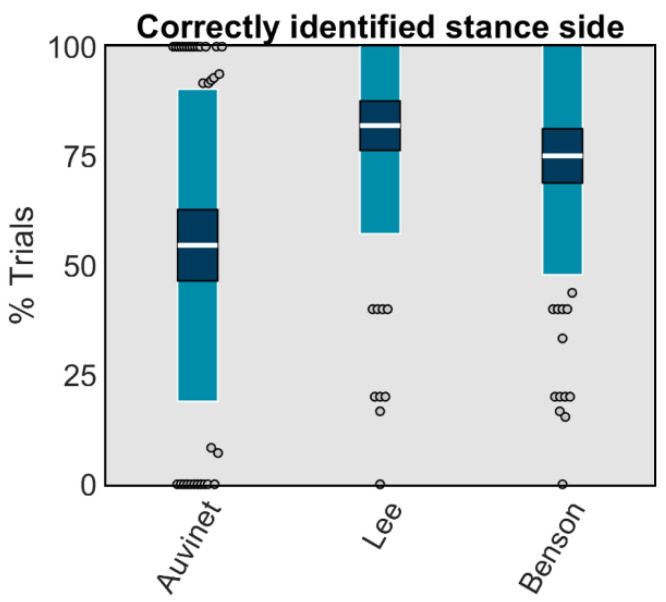
The white horizontal line represents the mean percentage of trials per participant where the side was correctly identified; dark blue represents ±95% confidence interval (±1.96 SEM) around the mean; and light blue represents ±1 SD around the mean. Gray dots represent participants outside ±1 SD.

**Figure 6 sensors-23-05022-f006:**
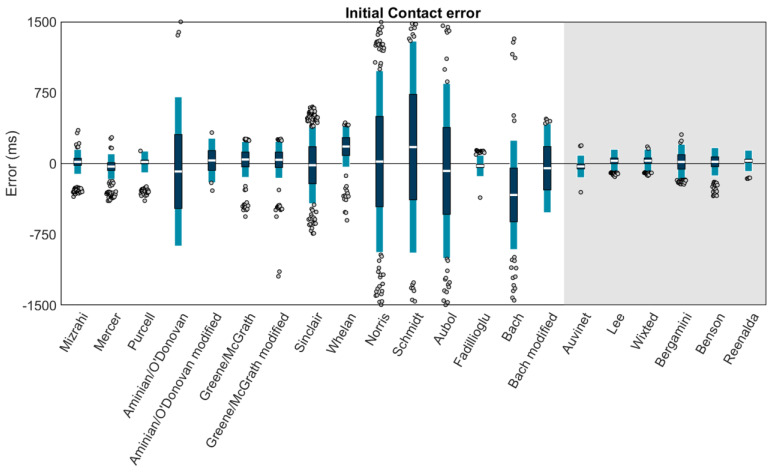
Means (white bar), ±1 within-method SD (dark blue), and ±95% LOA (1.96 SD of errors; light blue) for the IC estimation of each method. Gray dots represent trials falling outside the 95% LOA. A value of 0 indicates perfect agreement with the ground truth. Positive values indicate the IC was estimated later than the ground truth (after the force plate IC). Negative values indicate the IC was estimated earlier than the ground truth (before the force plate IC). Methods with a white background are for wearables on the shank. The methods on the gray background are for wearables on the sacrum/low back.

**Figure 7 sensors-23-05022-f007:**
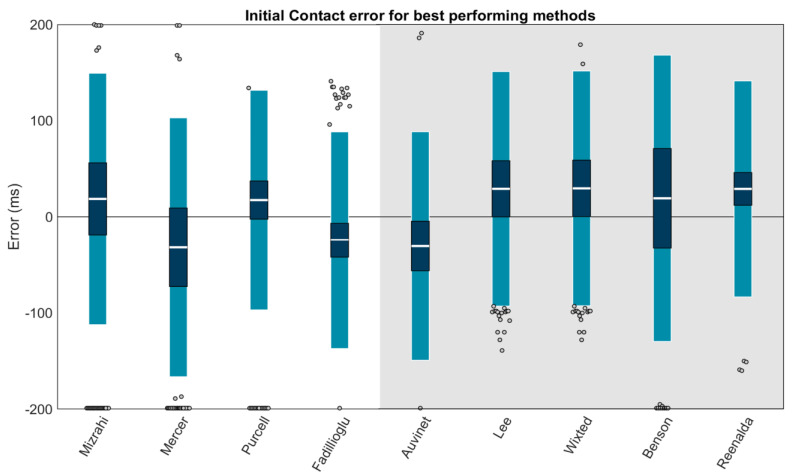
Only methods with LOAs within ±200 ms are plotted. Means (white bar), ±1 within-method SD (dark blue), and ±95% LOA (1.96 SD of errors; light blue) for the IC estimation of each method. Gray dots represent trials falling outside the 95% LOA. Outliers falling outside the ±200 ms range are plotted at ±200 ms. A value of 0 indicates perfect agreement with the ground truth. Positive values indicate the IC was estimated later than the ground truth (after the force plate IC). Negative values indicate the IC was estimated earlier than the ground truth (before the force plate IC). Methods with a white background are for wearables on the shank. The methods on the gray background are for wearables on the sacrum/low back.

**Figure 8 sensors-23-05022-f008:**
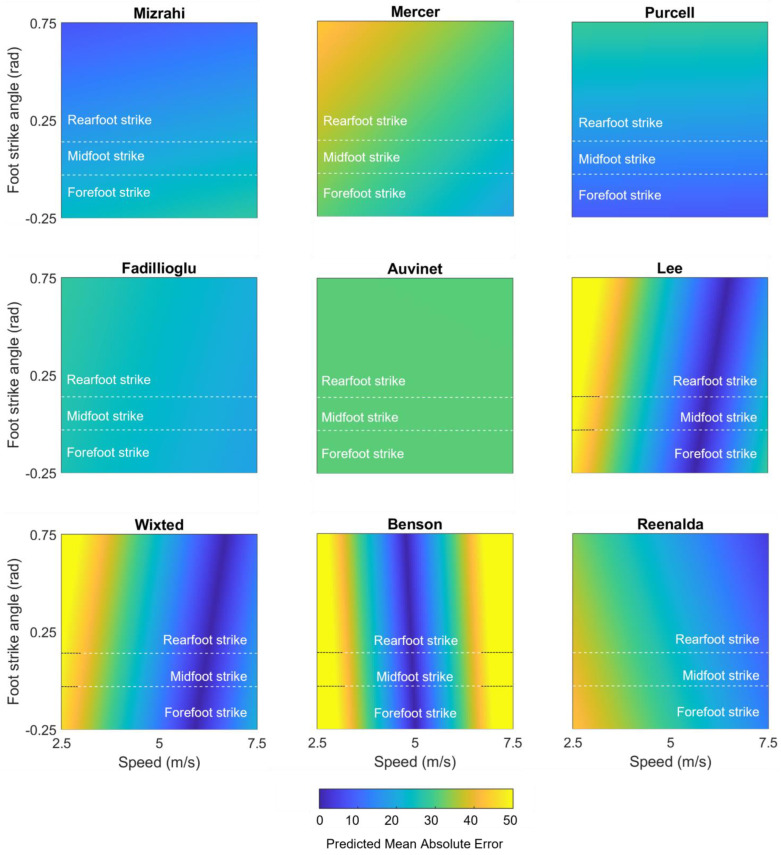
Mean absolute error in IC predicted by mixed effects models for each of the best-performing methods (LOAs within ±200 ms), plotted as a function of speed and foot strike angle. “Cooler” blue values represent lower predicted mean absolute errors, while “hotter” yellow values represent higher predicted mean absolute errors.

**Figure 9 sensors-23-05022-f009:**
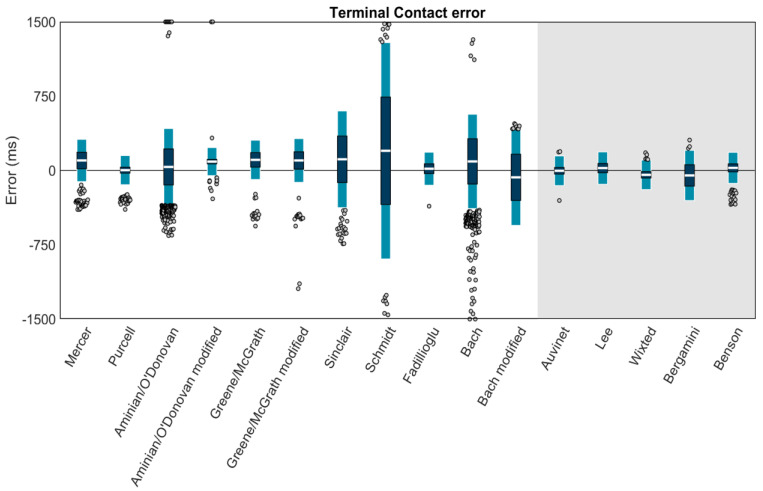
Means (white bar), ±1 within-method SD (dark blue), and ±95% LOA (1.96 SD of errors; light blue) for the TC estimation of each method. Gray dots represent trials falling outside the 95% LOA. A value of 0 indicates perfect agreement with the ground truth. Positive values indicate the TC was estimated later than the ground truth (after the force plate TC). Negative values indicate the TC was estimated earlier than the ground truth (before the force plate TC). Methods with a white background are for wearables on the shank. The methods on the gray background are for wearables on the sacrum/low back.

**Figure 10 sensors-23-05022-f010:**
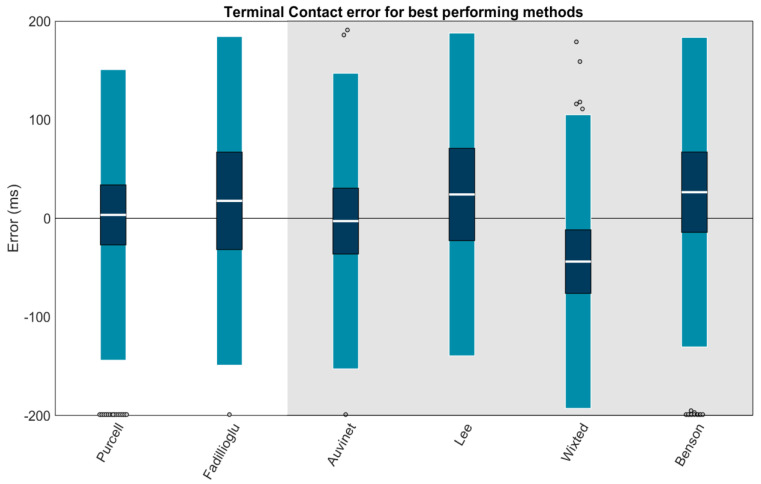
Only methods with LOAs within ±200 ms are plotted. Means (white bar), ±1 within-method SD (dark blue), and ±95% LOA (1.96 SD of errors; light blue) for the TC estimation of each method. Gray dots represent trials falling outside the 95% LOA. Outliers falling outside the ±200 ms range are plotted at ±200 ms. A value of 0 indicates perfect agreement with the ground truth. Positive values indicate the TC was estimated later than the ground truth (after the force plate TC). Negative values indicate the TC was estimated earlier than the ground truth (before the force plate TC). Methods with a white background are for wearables on the shank. The methods on the gray background are for wearables on the sacrum/low back.

**Figure 11 sensors-23-05022-f011:**
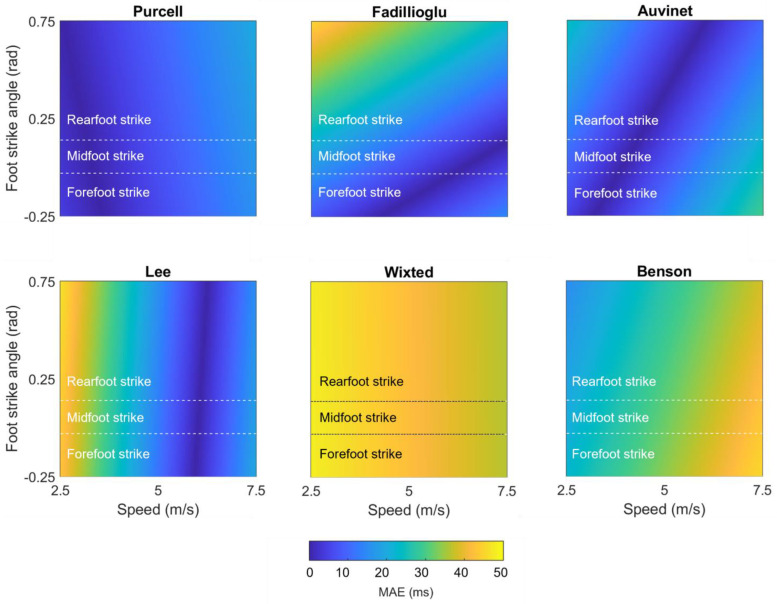
Mean absolute error in TC predicted by mixed effects models for each of the best-performing methods (LOAs within ±200 ms). Plotted as a function of speed and foot strike angle. “Cooler” blue values represent lower predicted mean absolute errors, while “hotter” yellow values represent higher predicted mean absolute errors.

**Figure 12 sensors-23-05022-f012:**
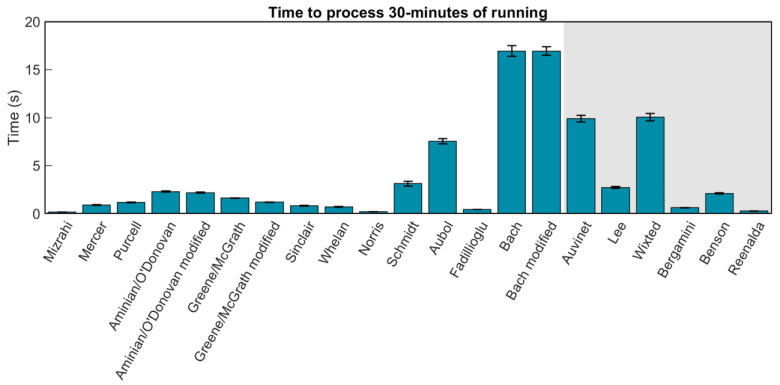
Magnitude of bars represents mean time to process 30 min of steady state running data sampled at 1000 Hz. Error bars represent ±1 SD about the mean.

**Table 2 sensors-23-05022-t002:** Effects of surface, running speed, and foot strike angle on initial contact (IC) and terminal contact (TC) estimation across methods. Numbers represent coefficients for the intercept of surface (added to model estimates for the floor condition but not the track condition) and the slopes of running speed and foot strike angle. * Significant (*p* ≤ 0.05) effects are highlighted in blue.

	Initial Contact (IC)			Terminal Contact (TC)		
Method	Surface	Speed	Foot Strike	Surface	Speed	Foot Strike
Mizrahi	−5.02	0.71	−15.91	n/a		
Mercer	−5.07	2.59	−11.99	−0.07	34.17 *	−47.56
Purcell	0.74	−0.14	19.66	2.46	4.00 *	4.02
Aminian/O’Donovan	2.81	−31.55	19.26	−15.10	11.29	−1.83
Aminian/O’Donovan modified	11.40	−5.58	−0.12	−1.26	−5.95 *	10.76
Greene/McGrath	2.22	−5.53	23.35	1.12	1.18	−16.43
Greene/McGrath modified	−1.91	−2.70	9.89	3.79	0.69	−25.87
Sinclair	−7.24	0.18	−44.17	−9.19	−15.65	43.08
Whelan	−14.22	−13.89 *	74.18 *	n/a		
Norris	−23.89	−36.41	148.44	n/a		
Schmidt	−11.17	209.61 *	49.38	−2.46	227.50 *	18.75
Aubol	−26.90	−3.94	34.33	n/a		
Fadillioglu	0.20	1.44	−2.13	1.27	−4.44	37.21
Bach	19.63	9.71	32.00	−7.44	−3.37	−44.60
Bach modified	−1.07	36.60 *	68.71	1.75	54.45 *	75.07
Auvinet	−3.08	0.01	0.13	0.15	6.86 *	−19.55
Lee	−1.03	−15.11 *	13.29	0.74	−12.46 *	4.05
Wixted	−2.99	−13.39 *	10.53	−1.52	2.37	0.20
Bergamini	−9.77	−19.31 *	12.63	−9.95	2.87	−34.54
Benson	0.11	−25.14 *	−6.34	0.46	4.60 *	−7.39
Reenalda	−0.13	−5.93 *	−9.39	n/a		

## Data Availability

Data are not publicly available due to stipulations in our IRB protocol, however, all software has been made publicly available at https://github.com/DovinKiernan/REID_IMU_Running_Event_ID published on 3 May 2023.

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
