# Peer review of "Unsupervised Gait Event Identification with a Single Wearable Accelerometer and/or Gyroscope: A Comparison of Methods across Running Speeds, Surfaces, and Foot Strike Patterns"

_sensors, 2023, doi:10.3390/s23115022_

Round 1
Reviewer 1 Report
Good research contribution is provided by authors but I wonder how readers will have interest in this work. Only those readers who are working on this area can concentrate because this.
(1) More pages are incorporated as a Supplementary Section.
(2) Overall references are 74 only whereas this paper is having more than 60 pages.
(3) Is it possible for authors to make it more concise? I think 40 pages are good enough.
Author Response
Good research contribution is provided by authors but I wonder how readers will have interest in this work. Only those readers who are working on this area can concentrate because this.
We believe that the volume of work already conducted in this area (e.g., the 18 candidate methods examined here) and the interest that this work has received (an average of 124 citations per candidate method on Google Scholar) demonstrates the current interest in identifying running Initial and Terminal Contact (IC and TC) events with wearables. Further, we believe that—if in-field biomechanics methods such as these can be validated—it will open new possibilities for the collection of ecologically valid, in-field biomechanics, expanding both the volume and quality of data available to biomechanists.
(1) More pages are incorporated as a Supplementary Section.
We appreciate the reviewer’s concern with the length of the paper (19 pages) and have tried to tighten it wherever possible (particularly in condensing the Discussion and Supplement). We are, however, trying to strike a balance between these concerns and the concerns of other reviewers who requested additional details be included in the main paper. We believe that, with the changes we’ve implemented, the current paper strikes a balance between providing sufficient information to understand our methods and results without providing erroneous detail.
(2) Overall references are 74 only whereas this paper is having more than 60 pages.
Thank you for the recommendation. We have added over a dozen citations and believe that we have been thorough in identifying and citing relevant literature. When considering the number of references please note that the main paper is only 19 pages (including many figures) and has 90 references, while the Supplemental Materials (which simply detail the individual methods and their results and thus do not require many additional references) make up the rest of the page count.
(3) Is it possible for authors to make it more concise? I think 40 pages are good enough.
We appreciate the reviewer’s concern with the length of the paper (19 pages) and have tried to tighten it wherever possible (particularly in condensing the Discussion and the Supplement). The Supplemental Materials remain quite long; however, an important aim of this paper is to provide a reproducible set of methods and we believe these materials are critical to that effort. Without these materials readers interested in additional details would not have: (1) clarity about our interpretation and implementation of each method, (2) full results for each method, nor (3) sufficient detail to understand and reproduce our IMU data processing (which have been included to satisfy the concerns of other reviewers).

Reviewer 2 Report
This paper provides a comprehensive analysis that overcomes certain limitations of existing methods. The analysis presented in this paper can assist users in making informed decisions about which method to choose from among the many available options for their specific application. Overall, this paper is of high quality. However, there is still room for improvement in certain areas, recommend accepting after minor revisions.
I do have a few suggestions for improvement:
1. In order to improve the clarity of your experimental procedures, the experimental images that illustrate the process of participants wearing their own shoes and running on the embedded force plate is necessary. These images will help readers to better visualize the setup and experimental conditions, and thus improve their understanding of the study design and methodology.
2. In the data processing section, it is recommended to provide a more detailed mathematical description of the processing steps.
3. It is recommended to include a detailed description of the coordinate system transformation process for the IMU during the experimental procedure. Additionally, it would be beneficial to provide a comparison between the raw and processed data before and after filtering and coordinate system transformation. This will aid in the understanding of the specific data processing steps and their effects on the data.
4. The analysis presented in the paper is comprehensive. However, the conclusion section appears verbose, suggests providing concrete recommendations on how to apply the methods in specific scenarios using figures and tables, which including factors such as IMU placement, speed, surface conditions, and others."
Author Response
This paper provides a comprehensive analysis that overcomes certain limitations of existing methods. The analysis presented in this paper can assist users in making informed decisions about which method to choose from among the many available options for their specific application. Overall, this paper is of high quality. However, there is still room for improvement in certain areas, recommend accepting after minor revisions.
I do have a few suggestions for improvement:
- In order to improve the clarity of your experimental procedures, the experimental images that illustrate the process of participants wearing their own shoes and running on the embedded force plate is necessary. These images will help readers to better visualize the setup and experimental conditions, and thus improve their understanding of the study design and methodology.
Thank you for this recommendation. To improve clarity, we added a photo to Figure 3 that shows a participant running in their own shoes and contacting the embedded force plate under the track surface. The photo also captures the position of the speed gates, conventional camera, and shank IMUs.
- In the data processing section, it is recommended to provide a more detailed mathematical description of the processing steps.
Thank you for the recommendation. An important goal of this paper is reproducibility; thus, we have added a section to the Supplemental Material better detailing our IMU data processing. This section either details the processing/math we used (if novel) or better describes and links to the original sources from which our methods were derived. We believe that these additions provide interested readers enough detail to fully understand, evaluate, and reproduce our IMU data processing.
- It is recommended to include a detailed description of the coordinate system transformation process for the IMU during the experimental procedure. Additionally, it would be beneficial to provide a comparison between the raw and processed data before and after filtering and coordinate system transformation. This will aid in the understanding of the specific data processing steps and their effects on the data.
Again, thank you for pointing out this lack of clarity in our methods. An IMU data processing section was added to the end of the Supplemental Material. All three coordinate transformations are fully described (segment, wearable, and a pseudo-global “tilt-corrected”) and example data are provided for each step of IMU data processing.
- The analysis presented in the paper is comprehensive. However, the conclusion section appears verbose, suggests providing concrete recommendations on how to apply the methods in specific scenarios using figures and tables, which including factors such as IMU placement, speed, surface conditions, and others."
We have attempted to tighten the paper and be more concise (particularly in the Discussion). We have also added a Conclusions section to underscore concrete recommendations and future directions.

Reviewer 3 Report
In this work, the authors have examined 18 techniques that can be used to detect the beginning and end of each step for 74 runners using data from a single wearable sensor placed on the shank or sacrum by varying foot strike angles, surfaces, and speeds. They recommended the Purcell or/and Fadillioglu methods and the Auvinet or/and Reenalda methods when the wearable sensor was on the shank and sacrum, respectively. Lastly, they suggested the Lee method, which has an accuracy of 81.95% for foot contact identification. The contributions of the work are original, engaging, and motivating. A set of well-attributed numerical results are given in the manuscript. However, a few critical concerns need to be addressed to improve the quality of the manuscript.
1. In addition to Table 1, the authors are suggested to highlight the limitations of existing methods in the text with an appropriate set of references. The description on page 3, "Most methods were developed and validated.....participants." should be described explicitly with different candidate methods from Table 1.
2. Moreover, the authors are suggested to highlight the difference in the methodology adopted by different studies considered in this work. Such details must be briefly moved from supplementary material to the main manuscript.
3. The authors are suggested to include the demographics and inclusion/exclusion criteria of the participants included in this study. Surprisingly, no actual pictures of humans with running trials are given in the manuscript. At least one sample picture should be mounted on a human participant with a hardware setup.
4. There should be more descriptions of Figures 4 and 5. On page 8, the authors write, ' The accuracy of these methods is plotted in Fig. 5.'; however, the y-axis of Figure 5 shows percent trials again like Figure 4.
5. The description of marked observations in Table 2 is missing in the manuscript. The authors have directly mentioned the numerical results without any reasoning or explanations. While comparing with different candidate methods, how do the authors ensure that the hardware/positioning calibrations of the wearable sensor remain the same?
6. The authors further suggested including explanations for the heatmap (Figures 8 and 11) to improve the manuscript's readability. It is impractical to interpret such plots by seeing them. The critical arguments need to be highlighted.
7. A conclusion should summarize the present work's critical observations and findings. Moreover, what are the limitations and future scope of the work? Do the authors imply that there may be no further need for new methods to detect foot events of runners?
Author Response
In this work, the authors have examined 18 techniques that can be used to detect the beginning and end of each step for 74 runners using data from a single wearable sensor placed on the shank or sacrum by varying foot strike angles, surfaces, and speeds. They recommended the Purcell or/and Fadillioglu methods and the Auvinet or/and Reenalda methods when the wearable sensor was on the shank and sacrum, respectively. Lastly, they suggested the Lee method, which has an accuracy of 81.95% for foot contact identification. The contributions of the work are original, engaging, and motivating. A set of well-attributed numerical results are given in the manuscript. However, a few critical concerns need to be addressed to improve the quality of the manuscript.
- In addition to Table 1, the authors are suggested to highlight the limitations of existing methods in the text with an appropriate set of references. The description on page 3, "Most methods were developed and validated.....participants." should be described explicitly with different candidate methods from Table 1.
Thank you for the recommendation. In the paragraph describing the limitations faced by existing methods (5th paragraph of the Introduction) we now explicitly list the candidate methods that face each limitation.
- Moreover, the authors are suggested to highlight the difference in the methodology adopted by different studies considered in this work. Such details must be briefly moved from supplementary material to the main manuscript.
To satisfy this recommendation at the end of page 2/beginning of page 3 we now describe which signals are used by the candidate methods and how they are used. To balance other reviewers’ concerns regarding length, this overview remains brief but interested readers can find full details regarding our interpretation and implementation of each method in the Supplemental Materials.
- The authors are suggested to include the demographics and inclusion/exclusion criteria of the participants included in this study. Surprisingly, no actual pictures of humans with running trials are given in the manuscript. At least one sample picture should be mounted on a human participant with a hardware setup.
Thank you for pointing out this omission, inclusion criteria have been added to the beginning of the Participants section.
To improve clarity, we have also included a sample picture of a participant running in their own shoes and contacting the embedded force plate under the track surface. The picture also captures the position of the speed gates, conventional camera, and shank IMUs. For the IMU fixation system, however, we believe the included cartoon better captures the details as all equipment is black (black IMUs on black Velcro on black belts with black silicone) making it very difficult to parse from a photo. Further, given the bilateral mounting of the IMUs, it is difficult to capture the fixation location of all IMUs in a single photo. We experimented with both a photo and a cartoon and found these details much easier to visualize with the cartoon.
- There should be more descriptions of Figures 4 and 5. On page 8, the authors write, ' The accuracy of these methods is plotted in Fig. 5.'; however, the y-axis of Figure 5 shows percent trials again like Figure 4.
Thank you for pointing out this lack of clarity. Additional in-text description has been added to Figure 4 explaining that it describes the percentage of trials where no gait events could be identified in the running data analyzed for that trial. For Figure 5 we improved understandability by showing the inverse results (successful ID instead of unsuccessful) and have added additional text explaining that it reports the percentage of trials that correctly identified the foot that was in contact with the ground during stance for the three sacrum methods capable of identifying the foot in contact with the ground. Results for both are now also more fully described.
- The description of marked observations in Table 2 is missing in the manuscript. The authors have directly mentioned the numerical results without any reasoning or explanations. While comparing with different candidate methods, how do the authors ensure that the hardware/positioning calibrations of the wearable sensor remain the same?
Again, thank you for pointing out the lack of clarity. We have added additional details describing that Table 2 is derived from our second set of linear mixed effects models that added surface, speed, and foot strike angle as potential explanatory variables for each candidate method. We now better explain that the values in the Table are the coefficients for each term in those models and that the bolding/color corresponds to whether the explanatory variable explained a significant amount of the model’s error.
We also added text to clarify that the data entered into each candidate method is identical, coming from the same 642 trials (method 1, …, 21 are each applied to trial 1, …, 642 resulting in a total of 13,482 possible observations). This type of ‘replicate measurement’ is the recommended approach for comparison across methods (see: Bland & Altman, 1999; Myles, 2007; Carstensen, 2008). Using this approach, for a given trial, the IMU positioning does not vary across methods. Similarly, the calibrations are IMU-specific (simply ensuring accurate angular velocities and accelerations are expressed in an orthogonal coordinate system square to the IMU housing); thus, these do not vary trial-trial or method-method either.
- The authors further suggested including explanations for the heatmap (Figures 8 and 11) to improve the manuscript's readability. It is impractical to interpret such plots by seeing them. The critical arguments need to be highlighted.
Thanks for pointing out the lack of clarity here. We have provided additional explanations of the heat maps, describing that the “hotter” yellow colors correspond to higher predicted mean absolute errors while the “cooler” blue colors correspond to lower predicted mean absolute error. We have also added explanations that the x-axis corresponds to the running speeds while the y-axis corresponds to foot strike angle with the horizontal lines demarcating different foot strike categories (rear foot, mid foot, and forefoot). Finally, we have summarized the effects observed from these heat maps.
- A conclusion should summarize the present work's critical observations and findings. Moreover, what are the limitations and future scope of the work? Do the authors imply that there may be no further need for new methods to detect foot events of runners?
We have added a conclusions section to underscore our limitations, our recommendations for the use of current methods, and the development of future methods. Critically, we point out that (1) our results are limited to the conditions we used in this study and that future work should validate the recommended methods across other conditions, (2) the methods used here could likely be improved upon either by combining techniques, developing new techniques, or adding additional information (e.g., speed), and (3) that future methods should not be developed without comparison to existing methods.

Round 2
Reviewer 3 Report
The authors have addressed all the concerns raised by the reviewer. The manuscript can be accepted without any further revisions. Just a small suggestion, if possible, please submit the Supplementary material into a separate supplementary file.